# Cancer and Pregnancy: Estimates in Italy from Record-Linkage Procedures between Cancer Registries and the Hospital Discharge Database

**DOI:** 10.3390/cancers15174305

**Published:** 2023-08-28

**Authors:** Daniela Pierannunzio, Alice Maraschini, Tania Lopez, Serena Donati, Rosalba Amodio, Fortunato Bianconi, Rossella Bruni, Marine Castaing, Claudia Cirilli, Giovanna Fantaci, Linda Guarda, Silvia Iacovacci, Lucia Mangone, Guido Mazzoleni, Walter Mazzucco, Anna Melcarne, Elisabetta Merlo, Fabio Parazzini, Fedro Alessandro Peccatori, Massimo Rugge, Giuseppe Sampietro, Giovanni Scambia, Giovanna Scarfone, Ausilia Sferrazza, Fabrizio Stracci, Antonina Torrisi, Maria Francesca Vitale, Silvia Francisci

**Affiliations:** 1National Centre for Disease Prevention and Health Promotion, Italian National Institute of Health, 00162 Rome, Italy; 2Technical-Scientific Statistical Service, Italian National Institute of Health, 00162 Rome, Italy; 3Clinical Epidemiology and Cancer Registry Unit, Azienda Ospedaliera Universitaria Policlinico (AOUP), 90127 Palermo, Italy; 4Area Operativa ICT—PuntoZero Scarl, 06128 Perugia, Italy; 5AReSS Puglia, 70121 Bari, Italy; 6Siracusa Cancer Registry, Health Unit of Siracusa, 96100 Siracusa, Italy; 7Dipartimento di Epidemiologia e Comunicazione del Rischio, AUSL di Modena, 41126 Modena, Italy; 8UOS Registro Tumori ASP Trapani, 91100 Trapani, Italy; 9SC Osservatorio Epidemiologico ATS Valpadana, 46100 Mantova, Italy; 10RT LT, Dipartimento di Prevenzione ASL Latina, 04100 Latina, Italy; 11European Epidemiology Unit, Gynecologic Oncology Department, AUSL-IRCCS di Reggio Emilia, 42122 Reggio Emilia, Italy; 12Tumor Register South Tyrol, 39100 Bolzano, Italy; 13Registro Tumori ASL di Lecce, 73100 Lecce, Italy; 14UOC Epidemiologia, Agenzia per la Tutela della Salute (ATS) della Brianza, 20900 Monza, Italy; 15Dipartimento di Scienze Cliniche e di Comunità, Università degli Studi di Milano, 20122 Milano, Italy; 16Gynecologic Oncology Department, European Institute of Oncology IRCCS, 20141 Milan, Italy; 17Surgical Pathology & Cytopathology Unit, Department of Medicine (DIMED), University of Padova, 35128 Padova, Italy; 18Veneto Tumor Registry (RTV), Veneto Regional Authority, 35132 Padova, Italy; 19Health Protection Agency of Bergamo (ATS Bergamo), 24121 Bergamo, Italy; 20Fondazione Policlinico Universitario Agostino Gemelli IRCCS, Università Cattolica Roma, 00168 Roma, Italy; 21Gynecologic Oncology Unit, Fondazione IRCCS Cà Granda Ospedale Maggiore Policlinico, 20122 Milano, Italy; 22ASP Ragusa-Dipartimento Medico di Prevenzione, UOSD Registro Tumori, 97100 Ragusa, Italy; 23Associazione Nazionale dei Registri Tumori—AIRTUM—Public Health Section, Department of Medicine and Surgery, University of Perugia, 06132 Perugia, Italy; 24Registro Tumori Integrato, Azienda Ospedaliero Universitaria Policlinico “G. Rodolico—San Marco”, 95123 Catania, Italy; 25Napoli 3 Sud Cancer Registry, 80031 Brusciano, Italy

**Keywords:** cancer, pregnancy, pregnancy outcome, reproductive health, childbearing age, record linkage, cohort study, population-based data, cancer registry records, hospital discharge data

## Abstract

**Simple Summary:**

Concurrence of pregnancy and cancer diagnosis is an uncommon but not rare event: about 1 pregnancy-associated cancer (PAC) out of 1000 pregnancies is the estimation currently available. This frequency is growing due to postponing childbearing to age groups more at risk of cancer. Interest in this topic is both epidemiological and clinical: improvement of diagnostic and therapeutic techniques makes management of cancer increasingly compatible with pregnancy. The occurrence of PAC challenges women and clinicians who need to manage the two events, safeguarding fetal outcomes without changing the maternal prognosis. This retrospective study aims to provide estimates for PAC and its time trend in Italy by analyzing cross-referenced data from population-based cancer registries and hospital discharges. The proposed methodology is applicable to other populations with available data from Cancer Registries linkable at an individual level with hospitalizations.

**Abstract:**

The aim of this study is to describe the frequency and trend of pregnancy-associated cancer (PAC) in Italy, an increasingly relevant phenomenon due to postponing age at childbirth. To this purpose, a population-based retrospective longitudinal study design based on cohorts of women aged 15–49 diagnosed with cancer and concomitant pregnancy is proposed. The study uses 19 population-based Cancer Registries, covering about 22% of Italy, and linked at an individual level with Hospital Discharge Records. A total of 2,861,437 pregnancies and 3559 PAC are identified from 74,165 women of the cohort with a rate of 1.24 PAC per 1000 pregnancies. The most frequent cancer site is breast (24.3%), followed by thyroid (23.9%) and melanoma (14.3%). The most frequent outcome is delivery (53.1%), followed by voluntary termination of pregnancy and spontaneous abortion (both 12.0%). The trend of PAC increased from 2003 to 2015, especially when the outcome is delivery, thus confirming a new attitude of clinicians to manage cancer throughout pregnancy. This represents the first attempt in Italy to describe PAC from Cancer Registries data; the methodology is applicable to other areas with the same data availability. Evidence from this study is addressed to clinicians for improving clinical management of women with PAC.

## 1. Introduction

Due to the postponing of parenthood and the increase in maternal age at birth, the frequency of pregnant women with cancer diagnoses has been rising in recent decades [1]. Cancer and pregnancy concurrence is challenging for clinicians required to preserve mother and fetal health, for women who need to be supported facing two crucial events simultaneously, and for epidemiologists aiming at measuring and describing this infrequent and increasingly relevant phenomenon.

For many years, oncological treatments were considered incompatible with pregnancy, and induced abortion or preterm labor were the only options to avoid the delay of cancer treatments. To date, the available evidence shows that it is often possible to safeguard fetal outcomes without changing the maternal prognosis [2,3].

International literature investigating the occurrence of Pregnancy-Associated Cancer (PAC) varies according to the following aspects: (1) the clinical versus the population-based perspective [4,5]; (2) the type of data sources available to measure and monitor PAC, i.e., population-based registries (cancer registries and/or birth registries), hospital discharges, clinical records [6,7,8]; (3) the definition used for PAC, in particular, the event under study (only pregnancy or also its outcomes) [9,10] and (4) the length of the time window used to identify PAC (i.e., the number of months preceding and/or following pregnancy or cancer diagnosis) [4]. A complete overview is available in a recent systematic review [11].

The most common way to measure the frequency of PAC is in terms of a ratio of the number of cancers during pregnancy to the total number of pregnancies. About one PAC every 1000 pregnancies or pregnancy outcomes is the average value estimated among different studies, corresponding to about 8 cancer cases yearly per 100,000, thus indicating an event not frequent but still above the threshold used to identify rare cancers (defined as those with an annual incidence of less than six per 100,000 people in Europe) [12]. 

North European countries, together with the USA and Australia, count on a tradition in epidemiological studies related to this topic [6,10,13]: there was an increasing trend in the rate of PAC in these areas. A recent Swedish study using cancer registry data linked with hospital discharges reported that cutaneous malignant melanoma, breast, cervical, thyroid, and central nervous system were the most common sites of PAC during 1970–2018 [14].

Few studies come from Southern Europe. Clinician groups led population-based studies in Italy, identifying PAC through Hospital Discharge Records [8,15,16]. They measured a PAC frequency between 127.1 and 134.1 per 100,000 pregnancies; no trend in risk was observed within the calendar year.

The present study aims to measure and describe the frequency as well as the time trend of PAC according to the possible outcomes of pregnancy in Italy using population-based Cancer Registries linked to the individual level of Hospital Discharge Records.

## 2. Materials and Methods

A longitudinal retrospective study on cohorts of women of childbearing age (15–49 years) was carried out. It was a population-based study where the referring population was constituted by newly cancers’ diagnosed women. Incident cancers were identified by the Cancer Registries and linked with the Hospital Discharge Records using an anonymized unique identification code. 

Cancer Registries collect data on all cancer diagnoses occurring in every person residing in the area covered by cancer registration. In this study, the following information for each patient is provided: date of birth, date of diagnosis, gender, vital status, site of primary tumor, and morphology code. The data from population-based Cancer Registries are the basis for the estimation of the cancer burden and its trends over time and are crucial in the planning and evaluation of cancer control programs in the area of registration [17].

According to their data availability, 19 Cancer Registries participating in the study provided incidence cases in 2003–2015; a minimum of five years of incidence was required to participate. We included all women aged 15–49 years diagnosed with malignant cancer (all sites) between 2003 and 2015 in the populations covered by the 19 Cancer Registries participating in the study. We excluded benign, uncertain, and non-melanoma skin cancer.

All malignant cancers, excluding non-melanoma skin cancer, were analyzed with site and morphology coded according to the ICDO-3 classification (C00–C97 excluding C44 with morphology different from 8720–8790). In the case of multiple cancers, the earliest diagnosis was used in the analyses.

Hospital Discharge Records contain information at individual level from hospitals; each record refers to a single hospital admission and includes demographic information (date of birth, sex, place of birth, place of residence), clinical information (main discharge diagnosis and up to five secondary discharge diagnoses, main intervention/procedure and up to five secondary interventions/procedures coded according to International Classification of Diseases, 9th revision, Clinical Modification—ICD9-CM) and administrative information (coded according to the Diagnosis Related Groups—DRG coding system and dates of admission and discharge).

In order to identify PAC, obstetric hospitalizations occurring to women of the study cohorts in the period spanning from one year before to two years after the cancer diagnosis were selected. 

Obstetric hospitalizations are selected according to specific diagnostic and procedural codes (the full list of codes is reported in Appendix A). Sequences of similar obstetric events in the same woman are classified according to the final outcome in the following categories: miscarriage, voluntary termination of pregnancy, ectopic pregnancy, hydatidiform mole and birth. About 21% of obstetric hospitalizations cannot be classified according to a specific reproductive outcome and are categorized in a residual group as “other”.

Data at discharge is used as data of the obstetric event.

In order to identify the number of pregnancies, obstetric hospitalizations occurring in the same reference territories of the participating Cancer Registries and in the same years of incidence were selected; in case of multiple hospitalizations referred to the same woman and leading to one of the above-classified outcomes, she counts once.

Hospital Discharge Records do not include home births, but between 2006 and 2015, the average proportion of home births reported by the National Birth Register was 0.1%, a percentage that we consider negligible in terms of risk of underestimation [18].

Time trends of PAC by pregnancy outcome in 2003–2015 were investigated by log-linear models; possible significant changes, expressed as annual percentage changes, were assessed by the permutation test using the JoinPoint Regression Program [19].

The Istituto Superiore di Sanità (Italian National Institute of Health—ISS), in collaboration with the Italian Society of Gynaecology and Obstetrics (SIGO) and the Italian Cancer Registries Association (AIRTUM), coordinated the study.

The Ethics Committee of the ISS approved the protocol of the study.

## 3. Results

The 19 Cancer Registries involved in the study are located in the North (8), Centre (2), and South (9) of Italy, covering, on average, about 3 million women of childbearing age every year, generally representing 22.6% (ranging from a minimum of 18.7 in 2003 to a maximum of 25.8 in 2010) of the Italian female population aged 15–49; participating Cancer Registries and years of incidence are reported in Appendix A.

A total of 74,165 women aged 15–49 years were diagnosed with malignant cancer (all sites excluding non-melanoma skin cancer) between 2003 and 2015 (Table 1); 1329 (1.8%) had two or more cancers over the study period. Overall, 2,861,437 pregnancies were registered in the Hospital Discharge Records in 2003–2015 in the general female population aged 15–49 years. A total of 4657 obstetric hospitalizations occurred within the time window from one year before to two years after the cancer diagnosis, corresponding to 3559 women with PAC.

Among the 74,165 women aged 15–49 diagnosed with malignant cancer in 2003–2015 (Table 2), the most frequent cancer sites were breast (41.3%), thyroid and other endocrine glands (15.8%), and female genital organs (10.3%), with a median age at diagnosis of 43 years. Among the 3559 women with PAC, breast cancer ranked first (24.3%), followed by the thyroid and other endocrine glands (23.9%) and melanoma of the skin (14.3%), with a median age at diagnosis of 35 years. The last column of the table reports the *p*-value testing the difference in the two population proportions (women with cancer and women with PAC).

Obstetric hospitalizations were more common before cancer diagnosis (55%), and a peak in the number of obstetric admissions was registered in coincidence with diagnosis: 11.9% within 30 days of the day of diagnosis. (Figure 1). 

Over the entire period, for all cancers combined, the most frequent pregnancy outcome was birth (53%), followed by miscarriage (12%) and voluntary termination of pregnancy (12%) (Table 3). Birth, miscarriage, and ectopic pregnancy sharply fell the first year after diagnosis, while voluntary termination of pregnancy had a similar rate one year before and one year after diagnosis. Voluntary termination of pregnancy was most frequent one year before for all cancers and breast cancer and one year after thyroid diagnosis. For breast cancer, a significant reduction in all pregnancy outcomes, except ectopic pregnancy, was observed two years after diagnosis; in the same temporal window, obstetric events markedly rise when referred to thyroid diagnosis.

More than 20% of obstetric hospitalizations are in the category “other”.

Time trends of PAC by pregnancy outcome from 2003 to 2015 showed an increase in births and miscarriages and a decrease in voluntary termination of pregnancy (Figure 2). These trends were confirmed by the Joinpoint regression: for births, the annual percentage change was +7.8 in 2003–2015 (*p* < 0.05); for miscarriages the annual percentage change was +8.6 in 2003–2015 (*p* < 0.05) and for voluntary termination of pregnancy the annual percentage change was +2.2 up to 2012, and 18.8 thereafter (annual percentage change values not statistically significant); time trend of PAC for all outcomes combined was increasing with an annual percentage change value of 6.1 (*p* < 0.05).

On average, in the time period 2003–2015, the age trend of PAC for births increased sharply with age up to 30–34 years, remained almost stable at 35–39, and declined abruptly thereafter; for abortions and voluntary termination of pregnancy increased up to the age 35–39 age group and decreased thereafter (Figure 3). 

## 4. Discussion

Pregnancy-associated cancer is an uncommon event, very challenging for patients and clinicians. It requires early recognition and treatment, taking into account not only maternal health but also fetal safety. Pregnancy complicated by cancers should, therefore, always be referred to facilities with high-risk maternity units and multidisciplinary teams to favor the optimal diagnostic and therapeutic approach. Proper investigation of an underlying malignancy during pregnancy may be delayed due to pregnancy physiological changes that can affect pharmacokinetic parameters and possibly impact the effectiveness and security of pharmacological therapy [20,21,22,23,24]. According to the International Network on Cancer, Infertility and Pregnancy (INCIP) Registry data, iatrogenic preterm birth induced for maternal oncological reasons, rather than chemotherapy exposure, was the main cause of early postnatal complications and poor neonatal neurodevelopmental outcomes [5].

Despite the need for epidemiological data supporting clinical practice, uncertainties remain about the frequency and time trends of PAC. 

Our study contributes to the field by providing recent estimates and trends of the rate of PAC in Italy based on Cancer Registries data individually linked with Hospital Discharge Records, thus considering cancer diagnosis as a reference event and tracing obstetric hospitalizations one year before and two years after the cancer diagnosis in order to identify PAC.

The reason for selecting obstetric events up to one year before the cancer diagnosis, instead of 9 months, is related to the evidence that in clinical practice, the date of cancer diagnosis represents the end of a clinical path, with diagnostic tests carried out up to 12 months before the diagnosis is registered. Concerning the choice of considering two years after a cancer diagnosis, this seems reasonable for finalizing surgery and other first-course therapies.

The study found an overall rate of PAC of 1.24 per 1000 pregnancies, consistent with the literature as reported in a recent systematic review, with an average value of 1.09 PAC per 1000 pregnancies [11] and in other Italian studies reporting 1.34 PAC per 1000 pregnancies [8].

The increasing trend of the rate of PAC is also confirmed by other international studies [6,10,13].

Our study showed an increase in the rate of PAC for birth and miscarriage and a decrease in voluntary termination of pregnancy, thus confirming the current attitude to manage pregnancy and oncological treatment, and voluntary termination of pregnancy not being considered as the only option [7,25,26].

PAC were more frequent in the case of cancers of the breast, thyroid, and other endocrine glands; as concerning breast cancer, our finding is consistent with the literature, as concerning thyroid cancer, the high frequency is probably related to the diagnostic pressure in Italy [27]. Differences in the cancer ranking concerning the general female population aged 15–49 were also due to the different age composition of the two groups (women with PAC are younger); melanoma was the third cancer in the general female population, while it was the fifth among women with PAC [28].

Results by reproductive outcome differ by cancer site: with respect to all cancers, miscarriage and voluntary termination of pregnancy occurred more often in the case of breast cancer and birth in the case of thyroid cancer; this difference is probably related to the prognosis of these cancers and consequently less aggressive therapies. The study reported a peak in the number of obstetric admissions at the time of cancer diagnosis, probably due to the intensification of clinical monitoring for women facing a cancer diagnosis during pregnancy. Obstetric events 2 years after cancer diagnosis had a lower level, and the reduction was probably due to the women’s refusal to start pregnancy during or immediately after cancer treatments. On the other side, this period could be worthy of further investigation on fertility. The relation among pregnancy outcomes, cancer site, and distance from diagnosis could provide additional evidence for clinicians [29,30]. Additionally, a recent study reported that the timing of diagnosis impacts the clinical cancer characteristics, e.g., for breast cancer, thus influencing the therapeutic approach and survival at 5–10 years from pregnancy [31].

Younger women experience miscarriage and voluntary termination of pregnancy less frequently than women aged 35 years and over. The highest percentage of births detected among women aged 30–34 years is coherent with the Italian mean age at birth [18].

The most relevant strength of this study is the use of Cancer Registries’ to identify cancers. Cancer Registries ensure a high level of accuracy and completeness in case identification of detailed and reliable clinical information, thus allowing overcoming methodological limits related to the use of administrative health data only, as is the case for Hospital Discharge databases [32].

A second strength is the population coverage of Cancer Registries’ data, which guarantees that results refer to the whole population rather than to a clinically selected cohort. Moreover, our study design and methodology are applicable to other areas where Cancer Registries and Hospital Discharge Records are available. 

The main limitation of our study is related to the data availability and update: Hospital Discharge Records can be considered reliable and available at the national level starting from 2003, and Cancer Registries collect incidence retrospectively with an average delay of three years. Although not representative of Italy, the study cohort covers a high percentage of the national population, about 22%. 

Obstetric events not leading to hospitalization do not contribute to the identification of pregnancies in women with cancer, thus implying an underreporting of PAC in our results. The impact of this error seems negligible, given the consistency of our estimates with those reported in other cited studies based on data from surveillance systems of pregnancies and the low frequency of obstetric events managed in ambulatory only. 

Another limitation is the lack of high-resolution clinical variables, such as the TNM stage; this variable is not routinely collected by Cancer Registries but can be provided for a subset of patients, as is the case for women with PAC. As further development would be interesting to describe PAC by outcome and stage at diagnosis. In order to describe the clinical pathways for women with PAC, it would be necessary to integrate our dataset with information on outpatient healthcare services and drug use. In addition, it would be worth integrating more information on pregnancy (e.g., gestational age, relevant to estimate maternal survival and infant mortality [33]), delivery, and parental characteristics from the Certificate of Delivery Care Registry (CEDAP).

## 5. Conclusions

This study represents the first attempt in Italy to provide population-based estimates of pregnancies in women with cancer diagnosis by linking data from Cancer Registries with Hospital Discharge Records at an individual level. 

The findings of this study are directed at epidemiologists to measure the phenomenon and to clinicians to raise their awareness of the potential to treat cancer during pregnancy, of the need to properly counsel future parents confronted with a cancer diagnosis during pregnancy, and of the necessity to always consider referral to health facilities with knowledge and awareness about cancer treatment during pregnancy. Additionally, the methodology can be used in other regions or countries where Cancer Registries and Hospital Discharge Records data are available. A further direction of study could aim at producing evidence about survival perspectives for women with PAC.

## Figures and Tables

**Figure 1 cancers-15-04305-f001:**
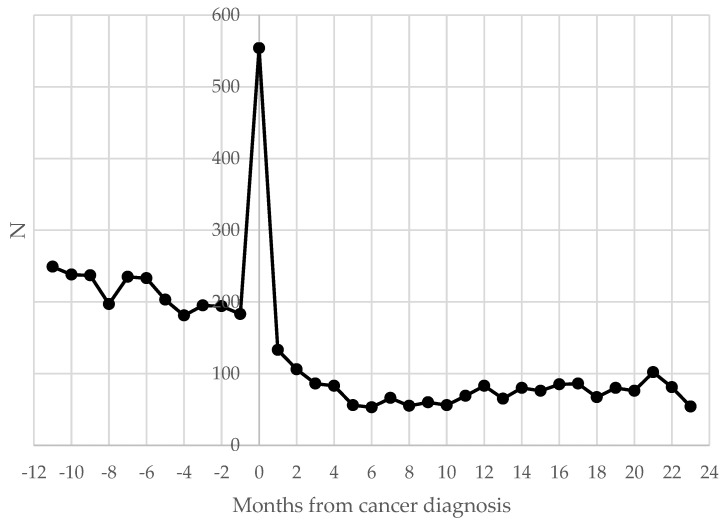
Distribution of obstetric hospitalizations (N) of women aged 15–49 by distance (in months) from cancer diagnosis: 2003–2015, Italy.

**Figure 2 cancers-15-04305-f002:**
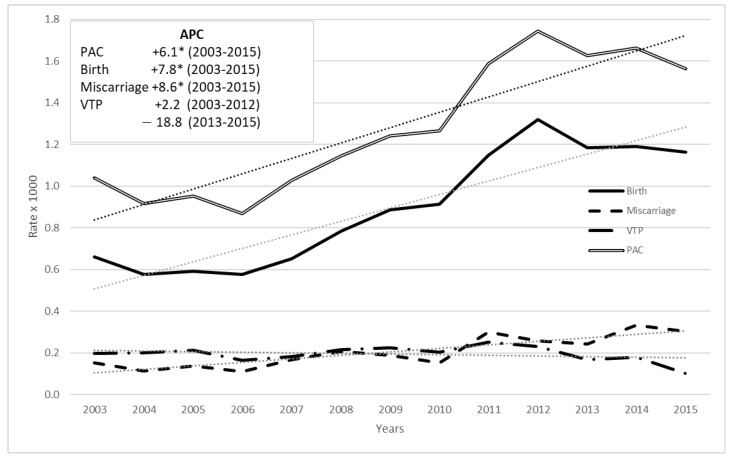
Time trend 2003–2015 of the rate of PAC (per 1000 pregnancies) among women aged 15–49 by pregnancy outcome: all cancers, Italy, (annual percentage change in box, linear trend in dots). ‘*’ stands for statistically significant APC values (*p* < 0.05).

**Figure 3 cancers-15-04305-f003:**
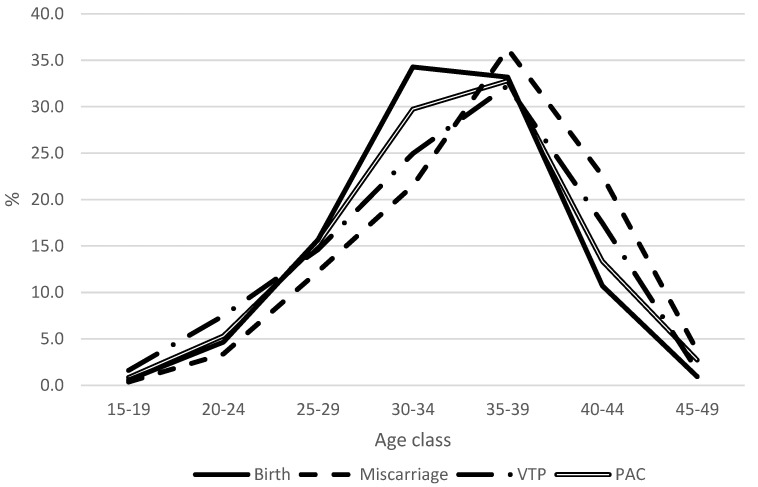
Age trend of PAC (%) by pregnancy outcome: all cancers, 2003–2015, Italy.

**Table 1 cancers-15-04305-t001:** Distribution of women with cancer diagnosis, number of pregnancies, number of women with PAC, and rate of PAC (per 1000 pregnancies) by year of cancer incidence: ages 15–49, all cancers, 2003–2015, Italy.

Year of Cancer Incidence	Women with Cancer Diagnosis	Pregnancies	PAC	PAC/Pregnancies × 1000
2003	4158	221,037	230	1.04
2004	4266	228,901	210	0.92
2005	4488	226,995	216	0.95
2006	4742	238,192	207	0.87
2007	5586	262,382	270	1.03
2008	6306	260,398	298	1.14
2009	6736	264,171	328	1.24
2010	6663	260,548	330	1.27
2011	6166	179,601	285	1.59
2012	6149	169,922	296	1.74
2013	6734	201,822	328	1.63
2014	6393	182,407	303	1.66
2015	5778	165,061	258	1.56
**2003–2015**	**74,165**	**2,861,437**	**3559**	**1.24**

**Table 2 cancers-15-04305-t002:** Distribution of women aged 15–49 and women with PAC aged 15–49 by cancer type (absolute and percent values): 2003–2015, Italy.

	Women 15–49		Women with PAC 15–49		
**Topography ***	**N**	**%**	**N**	**%**	** *p* **
Breast (C50)	30,626	41.29	866	24.33	0.0000
Thyroid and other endocrine glands (C73–C75)	11,731	15.82	849	23.86	0.0000
Femal genital organs (C51–C58)	7626	10.28	350	9.83	0.3893
Digestive organs (C15–C26)	5755	7.76	212	5.96	0.0001
Melanoma of the skin (C44 with morphology codes: 8720–8790)	5254	7.08	510	14.33	0.0000
Lymph nodes (C77)	2937	3.96	200	5.62	0.0000
Hematopoietic and reticuloendothelial systems (C42)	2806	3.78	169	4.75	0.0034
Respiratory system and intrathoracic organs (C30–C39)	2292	3.09	97	2.73	0.2179
Eye, brain, and other parts of central nervous system (C69–C72)	1408	1.90	98	2.75	0.0003
Urinary tract (C64–C68)	1361	1.84	85	2.39	0.0170
Lip, oral cavity, and pharynx (C00–C14)	876	1.18	41	1.15	0.8750
Connective, subcutaneous, and other soft tissues (C49)	503	0.68	35	0.98	0.0319
Unknown primary site (C80)	364	0.49	21	0.59	0.4100
Bones, joints, and articular cartilage of other and unspecified sites (C40–C41)	301	0.41	15	0.42	0.8863
Other and ill-defined sites (C76)	157	0.21	6	0.17	0.5829
Retroperitoneum and peritoneum (C48)	136	0.18	4	0.11	0.3293
Peripheral nerves and autonomic nervous system (C47)	32	0.04	1	0.03	0.6703
**Total**	**74,165**	**100.00**	**3559**	**100.00**	

* All morphologies were considered except for the Skin (C44), for which only the 8720–8790 codes were included.

**Table 3 cancers-15-04305-t003:** Distribution of women aged 15–49 with PAC by pregnancy outcome and timing (one year before cancer diagnosis, one year after cancer diagnosis, and two years after cancer diagnosis) for all cancers, breast cancer and thyroid and other endocrine glands cancer, % value (counts): 2003–2015.

	All Cancers			
Pregnancy Outcome	One Year before Cancer Diagnosis	One Year after Cancer Diagnosis	Two Years after Cancer Diagnosis	Total
Birth	59.7 (1477)	20.0 (495)	20.3 (502)	53.1 (2474)
Miscarriage	57.4 (323)	19.7 (111)	22.9 (129)	12.1 (563)
Voluntary termination of pregnancy	45.8 (257)	36.0 (202)	18.2 (102)	12.0 (561)
Ectopic pregnancy and hydatidiform mole	63.0 (46)	13.7 (10)	23.3 (17)	1.6 (73)
Pregnancy not classifiable	46.2 (456)	35.5 (350)	18.3 (180)	21.2 (986)
**Total**	**54.9 (2559)**	**25.1 (1168)**	**20.0 (930)**	**100.0 (4657)**
	**Breast**			
**Pregnancy Outcome**	**One Year before Cancer Diagnosis**	**One Year after** **Cancer Diagnosis**	**Two Years after** **Cancer Diagnosis**	**Total**
Birth	75.4 (419)	21.0 (117)	3.6 (20)	53.1 (556)
Miscarriage	77.6 (118)	17.1 (26)	5.3 (8)	14.5 (152)
Voluntary termination of pregnancy	61.4 (97)	28.5 (45)	10.1 (16)	15.1 (158)
Ectopic pregnancy and hydatidiform mole	63.6 (7)	9.1 (1)	27.3 (3)	1.1 (11)
Pregnancy not classifiable	50.6 (86)	44.1 (75)	5.3 (9)	16.2 (170)
**Total**	**69.4 (727)**	**25.2 (264)**	**5.4 (56)**	**100.0 (1047)**
	**Thyroid and Other Endocrine Glands Cancer**			
**Pregnancy Outcome**	**One Year before** **Cancer Diagnosis**	**One Year after** **Cancer Diagnosis**	**Two Years after** **Cancer Diagnosis**	**Total**
Birth	52.4 (326)	16.9 (105)	30.7 (191)	56.6 (622)
Miscarriage	45.1 (60)	21.8 (29)	33.1 (44)	12.1 (133)
Voluntary termination of pregnancy	36.3 (58)	42.5 (68)	21.2 (34)	14.6 (160)
Ectopic pregnancy and hydatidiform mole	66.7 (6)	0.0 (0)	33.3 (3)	0.8 (9)
Pregnancy not classifiable	39.7 (69)	24.1 (42)	36.2 (63)	15.8 (174)
**Total**	**47.3 (519)**	**22.2 (244)**	**30.5 (335)**	**100.0 (1098)**

## Data Availability

Data are unavailable due to privacy restrictions.

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
