# Peer review of "Cancer and Pregnancy: Estimates in Italy from Record-Linkage Procedures between Cancer Registries and the Hospital Discharge Database"

_cancers, 2023, doi:10.3390/cancers15174305_

Round 1

Reviewer 1 Report

The study provides additional and interesting data about pregnancy associated cancer. Results are stated clearly

Could there be an underrepresented rate of abortions? Or are the captured in the obstetric databases? I assume that not often there would be a need for hospitalization in case of abortion, whereas it would be the case in live-birth (or still birth). Furthermore, what is the average percentage of home-deliveries? If there is not yet a cancer diagnosis, there may not yet be an indication for hopitalization when giving birth

page 3, section MM, 13th line after MM. minor typo "all malignant ancers".... 

some minor typo's, see above. 

Author Response

The study provides additional and interesting data about pregnancy associated cancer. Results are stated clearly

Point 1: Could there be an underrepresented rate of abortions? Or are the captured in the obstetric databases? I assume that not often there would be a need for hospitalization in case of abortion, whereas it would be the case in live-birth (or still birth).

The study intercepts obstetric events through the hospital discharge records. Therefore, as correctly suggested by the reviewer, it does not capture all miscarriage events, especially early miscarriages that do not require hospitalization. However, this underestimation of miscarriages occurs among women with cancer and the general population, thus making the numerator and denominator included in the rate and the annual trend rate comparable.

Point 2: Furthermore, what is the average percentage of home-deliveries?

We thank the reviewer for the request and have added the following sentence to the manuscript to provide information for readers:

Hospital Discharge Records do not include home births but, between 2006 and 2015, the average proportion of home births reported by the National Birth Register was 0.1%, a percentage that we consider negligible in terms of risk of underestimation.

Point 3:If there is not yet a cancer diagnosis, there may not yet be an indication for hopitalization when giving birth

The number of home births in Italy is very small (as reported in Point 2) and therefore negligible in both women who have a cancer diagnosis and those who do not at the time of delivery.

page 3, section MM, 13th line after MM. minor typo "all malignant ancers".... 

Comments on the Quality of English Language

some minor typo's, see above. 

We thank the reviewer for the comment, we corrected the text

Reviewer 2 Report

·       The abstract should state briefly the purpose of the research, the principal results and major conclusions. An abstract is often presented separately from the article, so it must be able to stand alone

-          discuss the research aims, research gap and discuss the paper layout Add up-to-date references to support your discussion

Methods

·       The methodology of this study should be detailed, limit information was provided on method and materials.

·       Sampling strategy should be defined, how the samples were collected

·       The authors should be able to control/reduce  the selection bias

·       Sample size calculation is missing

·       internal and external validity of the study is totally missing

·       inclusion of and exclusion criteria is missing

Moderate editing of English language required

Author Response

The abstract should state briefly the purpose of the research, the principal results and major conclusions. An abstract is often presented separately from the article, so it must be able to stand alone.

Discuss the research aims, research gap and discuss the paper layout Add up-to-date references to support your discussion

The methodology of this study should be detailed, limit information was provided on method and materials

Abstract, Materials and Methods and Discussion have been revised accordingly.

Point 1: Sampling strategy should be defined, how the samples were collected

Response 1: We do not need a sampling strategy, since the study results are not based on a sample.

This is a longitudinal retrospective study using data from population-based Cancer registries linked at individual level with Hospital discharges database. Cancer Registries collect data on all cancer diagnoses occuring in every person residing in the area covered by cancer registration. According to this study design, all women with a malignant cancer diagnosis at the age 15-49 in the population covered by the 19 CR participating to the study is included in the study cohort.

This aspect is described in the materials and methods section and recalled in the discussion (page 9, line 4 from the top): A second strength is the population coverage of Cancer Registries’ data, which guarantees that results refer to the whole population rather than to a clinically selected cohort.

We modified the materials and methods section in order to clarify this point

Point 2: The authors should be able to control/reduce the selection bias

Response 2: According to our study design, we do not need to control/reduce the selection bias, since our results derive from the populations covered by the 19 Cancer Registries involved in the study.

However, there is an issue related to the representativeness of these 19 CRs with respect to the Italian National population. This issue is highlighted in the Discussion (page 9, line 5 from the top): “Although not representative of Italy, the study cohort covers a high percentage of the national population, about 22%”.

We modified the discussion in order to clarify this issue as a limitation of the study.

Point 3: Sample size calculation is missing

Response 3: see answer to question 1 above.

Point 4: internal and external validity of the study is totally missing

Response 4: Internal and external validity refers to how well the results among the study participants represent true findings and to which extent these results are generalizable to patients in daily practice, especially for the population that the sample is thought to represent. This concept of validity applies to all types of clinical studies, however it does not apply to our population study design.

Point 5: inclusion of and exclusion criteria is missing

Response 5: Inclusion/exclusion criteria are specified in the Materials and methods section:

As concerning the study cohort, we included all women aged 15-49 years diagnosed with malignant cancer (all sites excluding non-melanoma skin cancer) between 2003 and 2015 in the populations covered by the 19 Cancer Registries participating to the study. According to their data availability, each Cancer Registry provided incident cases for a minimum of five years in 2003-2015. All malignant cancers, excluding non-melanoma skin cancer, were analysed. In case of multiple tumours, the earliest diagnosis was used in the analyses.

As concerning pregnancies, we identify pregnancies of women from the study cohort by selecting obstetric hospitalizations, among those hospitalizations occurring within the time window from one year before to two years after the cancer diagnosis, according to specific diagnostic and procedural codes listed in Appendix A.

We modified the materials and methods section in order to clarify this point

Reviewer 3 Report

The Manuscript ID cancers-2462582 is relevant and makes contributions to science. In the Introduction, the object of study is presented with the various aspects of Pregnancy-Associated Cancer (PAC). The objective of the manuscript is to measure and describe the frequency as well as the time trend of PAC according to the possible outcomes of pregnancy in Italy using population based Cancer Registries linked to the individual level of Hospital Discharge Records. The methodology is well described and the results are presented in tables and graphs that are easy to understand.

It is suggested that there is a need for greater discussion of the results obtained in the research presented in the Manuscript ID cancers-2462582, highlighting the interpretation of the data expressed in the tables and graphs. The Conclusion could be directed towards answering the objective of the study.

Author Response

Point 1: It is suggested that there is a need for greater discussion of the results obtained in the research presented in the Manuscript ID cancers-2462582, highlighting the interpretation of the data expressed in the tables and graphs.

Response 1: We thank the reviewer for the suggestion. Comments on results and Discussion have been integrated and extended.

Point 2: The Conclusion could be directed towards answering the objective of the study

Response 2: Considering the aim of the study being “to measure and describe the frequency as well as the time trend of PAC according to the possible outcomes of pregnancy in Italy using population-based Cancer Registries linked to the individual level of Hospital Discharge Records” two sentences have been added in Conclusion to better detail the message for clinicians and to underline the relevance of survival analisys for further studies.

Reviewer 4 Report

Pierannunzio et al. conducted a study on pregnancy-associated cancer in Italian population. The authors used 19 cancer registries that covers about 22% of Italy. This data was linked with individual level information for hospital discharge records. The authors used joinpoint regression program to analyze the data and presented time and age trend data and other relevant descriptive data. This is an important study using nearly 3 million pregnancy data. The study could be improved by providing more data associated with PAC. The following are the comments to improve this manuscript.

Why didn’t the authors perform statistical test to compare the cancer type among control and PAC groups? This would have helped to understand which cancer types are significantly different between the groups despite the difference in mean ages.

Are there any data about the impact of PAC on babies?

Are there any data on pre-existing health issues such as diabetes, cardiovascular disease available from the registry to report for the PAC group?

Include the total number of pregnancies screened to the abstract section to improve the visibility and significance of this study.

Why did the authors prefer mean instead of median to report age data?

The authors mentioned “population-based CRs” in the method section. I believe the CR refers to cancer registries. The authors can either provide the abbreviation with expansion at the first instance and continue with the abbreviation through out the manuscript or remove it to maintain consistency.

There were two commas for pregnancies data for 2011 and none for 2012 in table 1.

Ectopic was written as etopic and hydatidiform was written as hydatiform in table 3.

In the discussion, the authors mentioned “recent systematic review” but did not provide the citation to know which study they were referring.

The study recommendations or message to clinicians is not clearly elaborated apart from simple mentioning of it.

If possible, the authors can provide a figure similar to figure 3 for pregnancy without cancer to know whether delayed parenthood has any association with PAC.

There are some grammatical errors that can fixed to improve the manuscript.
